# Surrogate Selection for Foot-and-Mouth Disease Virus in Disinfectant Efficacy Tests by Simultaneous Comparison of Bacteriophage MS2 and Bovine Enterovirus Type 1

**DOI:** 10.3390/v14122590

**Published:** 2022-11-22

**Authors:** Chae Hong Rhee, Seung-Chun Park, Moon Her, Wooseog Jeong

**Affiliations:** 1Veterinary Drugs & Biologics Division, Animal and Plant Quarantine Agency, 177 Hyeoksin 8-ro, Gimcheon-si 39660, Gyeongsangbuk-do, Republic of Korea; 2Laboratory of Veterinary Pharmacokinetics and Pharmacodynamics, College of Veterinary Medicine, Kyungpook National University, Daegu 41566, Gyeongbuk do, Republic of Korea

**Keywords:** food-and-mouth disease virus, disinfectant, bacteriophage MS2, bovine enterovirus type 1, surrogate virus, virucidal efficacy

## Abstract

In South Korea, testing disinfectants against foot-and-mouth disease virus (FMDV) that are contagious in livestock or that require special attention with respect to public hygiene can be manipulated only in high-level containment laboratories, which are not easily available. This causes difficulties in the approval procedure for disinfectants, such as a prolonged testing period. Additionally, the required biosafety level (BSL) in the case of FMDV has hindered its extensive studies. However, this drawback can be circumvented by using a surrogate virus to improve the performance of the efficacy testing procedure for disinfectants. Therefore, we studied bacteriophage MS2 (MS2) and bovine enterovirus type 1 (ECBO) with respect to disinfectant susceptibility for selecting a surrogate for FMDV according to the Animal and Plant Quarantine Agency (APQA) guidelines for efficacy testing of veterinary disinfectants. Effective concentrations of the active substances in disinfectants (potassium peroxymonosulfate, sodium dichloroisocyanurate, malic acid, citric acid, glutaraldehyde, and benzalkonium chloride) against FMDV, MS2, and ECBO were compared and, efficacies of eight APQA-listed commercial disinfectants used against FMDV were examined. The infectivity of FMDV and ECBO were confirmed by examination of cytopathic effects, and MS2 by plaque assay. The results reveal that the disinfectants are effective against MS2 and ECBO at higher concentrations than in FMDV, confirming their applicability as potential surrogates for FMDV in efficacy testing of veterinary disinfectants.

## 1. Introduction

Foot-and-mouth disease (FMD) is a severe, highly contagious viral disease of livestock, which has a significant economic impact [1]. The reported cost of control measures endured by the Korean government for five previous FMD epidemics from 2000 to 2011 was from USD 23.6 million to USD 1.9 billion, and for the 2014/2015 epidemic was USD 58.3 million in South Korea [2]. The disease causes severe production losses, and while the majority of affected animals recover, they become weakened and debilitated [1]. The FMD outbreak in South Korea resulted in a rapid decrease in the supply of meat products and a 24% increase in the consumer price, as compared to the same time period of the previous year [2].

The causative pathogen, foot-and-mouth disease virus (FMDV), is a small, non-enveloped, single-stranded RNA virus of the family *Picornaviridae*, genus *Aphthovirus* [3]. Generally, non-enveloped viruses are more stable outside their hosts and have a higher possibility of spreading via environmental agents [4,5]. Therefore, environmental control using the proper chemical disinfectants is important for the prevention and control of infectious diseases, along with vaccination, surveillance, and quarantine [6].

T5In South Korea, veterinary disinfectants are regulated by the Animal and Plant Quarantine Agency (APQA), and the agency currently has a system to review and register label claims of the formulations and recommended concentrations against the target microorganisms according to its guidelines for efficacy testing of veterinary disinfectants [7,8]. However, testing of disinfectants against FMDV can be conducted only in high-level containment laboratories, which are not easily available. This situation causes some difficulties in the approval procedure for disinfectants, including a prolonged testing period. Additionally, the required biosafety level (BSL) for FMDV has hindered its extensive studies. The APQA guidelines have approved the use of surrogate organism for claims of activity against bacteria [7]; however, this does not apply to viruses, and testing is required against each virus to be listed on the product label [8]. Therefore, the use of a surrogate for FMDV in chemical disinfectant testing has been suggested. We designed a study for the selection of a surrogate virus, which is susceptible to disinfectants and can help to improve the efficacy of their testing procedure. A surrogate has many characteristics which make it an ideal model for experimentation, such as easy to culture and handle, capable of reaching high virus titers, high levels of stability, etc. [8,9]. The major benefit of using a nonpathogenic surrogate is its requirement of a lower BSL than that for the actual pathogen [8]. In laboratory studies, the use of surrogates for disinfection testing and data based on testing with the surrogates are preferred while developing models of such highly infectious pathogens requiring studies on high-level containment [8,10,11,12]. This is because large-volume surrogate testing can be conducted and subsequently confirmed with small-volume infectious pathogen testing [13].

Studies on an appropriate surrogate have not been performed yet for disinfection testing with FMDV in South Korea. Therefore, the present study had two main objectives. First, we aimed to evaluate the virucidal efficacy of the representative active substances in commercial disinfectants used against FMDV by testing against FMDV and its surrogate candidates, bacteriophage MS2 (MS2) and bovine enterovirus type 1 (ECBO). Second, we aimed to confirm whether two surrogate candidates are appropriate and practical surrogates for virucidal efficacy testing. The virucidal efficacy of the representative active substances and commercial disinfectants against FMDV, MS2, and ECBO were tested according to the APQA guidelines. Finally, the observed effective concentrations with surrogate candidates were compared to determine their suitability as surrogate viruses for FMDV. All experiments using FMDV were performed in a BSL-3 facility, and those using MS2 and ECBO were performed in a BSL-2 facility in the APQA of South Korea.

## 2. Materials and Methods

### 2.1. Disinfectants

Six active substances that represent the main chemical groups of commercially available disinfectants were selected for the study. Their representative products, namely potassium peroxymonosulfate (PPMS; Sigma-Aldrich, St. Louis, MO, USA, CAS: 70693-62-8), sodium dichloroisocyanurate (NaDCC; Sigma-Aldrich, CAS: 2893-78-9), glutaraldehyde (GLT; Grade I, 25% in H_2_O; Sigma-Aldrich, CAS: 111-30-8), citric acid (CA; Sigma-Aldrich, CAS: 77-92-9), malic acid (MA; Sigma-Aldrich, CAS: 6915-15-7), and benzalkonium chloride (BZK; Sigma-Aldrich, CAS: 63449-41-2) were used. Additionally, ten APQA-approved commercial disinfectants currently used against FMDV, were also examined. All disinfectants were diluted in hard water (CureBio, Seoul, Korea) supplemented with 5% fetal bovine serum (FBS; Corning Inc., Corning, NY, USA) to obtain the specific concentrations that represent high-level organic soiling, as described in the APQA guidelines for efficacy testing of veterinary disinfectants [7,14]. To ensure maximum performance of the disinfectants, the dilutions were prepared immediately before use.

### 2.2. Host Cells

The LFBK and Vero cell lines were routinely maintained in Dulbecco’s modified Eagle’s medium (DMEM; Corning Inc., Corning, NY, USA) supplemented with 10% FBS (Corning Inc.) and 1% antibiotic-antimycotic solution (AA; Corning Inc.) and were used for these experiments. All cells were grown at 37 °C in a humid atmosphere with 5% CO_2_ saturated with water vapor. *Escherichia coli* (*E. coli*) C3000 (ATCC 15597) was purchased from the American Type Culture Collection (ATCC) and was grown as described in the study by Rhee et al. [15]. In brief, *E. coli* was cultivated in the culture broth while shaking it at 37 °C for 6 h and prepared to get a density of 0.2–0.3 at OD_600_. The culture broth was prepared to contain 10 g/L tryptone (BD Difco, Franklin Lakes, NJ, USA; Cat. 211705), 1 g/L yeast extract (BD Difco; Cat. 212750), 8 g/L sodium chloride (Sigma-Aldrich, St. Louis, MO, USA; Cat. 7647-14-5), 10 g/L glucose (Sigma-Aldrich; Cat. 50-99-7), 0.294 g/L calcium chloride (Sigma-Aldrich; Cat. 10043-52-4), and 0.01 g/L thiamine (Sigma-Aldrich; Cat. 67- 03-8).

### 2.3. Viruses

The FMDV used for the experiment was the O Jincheon strain. The MS2 and ECBO were purchased from ATCC. The FMDV was propagated in LFBK cells, and virus titration was performed after 3 d of incubation by microscopically determining the cytopathic effect (CPE) for FMDV-infected LFBK cells. ECBO, strain LCR-4 strain (ATCC VR-248), was propagated in Vero cells, and virus titration was performed after 5 d of incubation in the same manner for ECBO-infected Vero cells as described above. The coliphage MS2 (ATCC 15597-B1) was propagated and enumerated according to the Rhee et al. [15], with *E. coli* as host. Finally, the viruses were stored at −70 °C in 1 mL aliquots until further use.

### 2.4. Virucidal Efficacy Test

The virucidal efficacy tests of the disinfectants were performed in suspension, in accordance with the procedure described by the APQA guidelines for efficacy testing of veterinary disinfectants [7].

#### 2.4.1. Virus-Disinfectant Reaction

First, the viruses were mixed in the ratio 1:19 with hard water supplemented with 5% FBS to prepare the virus inoculum, which was subsequently mixed with each test concentration of the disinfectants in the ratio 1:1 and incubated at 4 °C for 30 min (obligatory condition) [7,14], while being vortexed every 10 min. For the virus control, hard water was used instead of the disinfectant solution. At the end of reaction time, test samples were immediately mixed with an equal amount of neutralizing medium of each test to quench the chemical activity of disinfectants: the FMDV and ECBO test samples with DMEM medium containing 10% FBS and 1% AA, and the MS2 test samples with culture broth containing 10% FBS. Subsequently, ten-fold serial dilutions (up to 10^−7^) were prepared in 2% FBS-supplemented medium of each virus for FMDV and ECBO, and culture broth for MS2. Virus titer of minimum 2 × 10^5^ TCID_50_/mL or PFU/mL was ensured in the virus control.

#### 2.4.2. Tissue Culture Infectious Dose Assay for FMDV and ECBO

Aliquots of 100 µL from each dilution of the virus-disinfectant mixtures were transferred into six wells of a 96-well microtiter plate, each containing 50 µL of LFBK or Vero cell cultures (1 × 10^6^ cells/mL), followed by incubation at 37 °C in a CO_2_ incubator. The treatment set-ups were observed every day, and microscopic examination confirmed the incidence of virus-induced CPE after 3 d for FMDV and 5 d for ECBO. The viral titer was calculated as 50% tissue culture infectious dose per mL (TCID_50_/mL) in log units using the Spearman–Karber method [16]. All experiments using FMDV were performed in a BSL-3 facility, and those using ECBO were performed in a BSL-2 facility in the APQA of South Korea.

#### 2.4.3. Double Agar Layer Plaque Assay for MS2

The assay was performed according to Rhee et al. [15]. In brief, aliquots of 50 µL from each dilution of the virus-disinfectant mixtures were added into a test tube with 5 mL of top agar (10 g/L tryptone, 1 g/L yeast extract, 8 g/L sodium chloride, 7 g/L agar (BD Difco, Franklin Lakes, NJ, USA; Cat. 214010), 10 g/L glucose, 0.294 g/L calcium chloride, and 0.01 g/L thiamine) and 10 µL of *E. coli* host at a density of 0.2–0.3 at OD_600_, mixed carefully, and kept in a dry heating block at 50 °C for 5 min. The mixture was poured on a bottom agar plate (10 g/L tryptone, 1 g/L yeast extract, 8 g/L sodium chloride, 15 g/L agar, 10 g/L glucose, 0.294 g/L calcium chloride, and 0.01 g/L thiamine) plate and allowed to solidify. The mixture was then incubated upside-down at 37 °C for 18 h, after which the MS2 plaques were counted. The viral titer was calculated as 50% plaque forming units per mL (PFU/mL) in log units using the Spearman–Karber method [16]. All experiments using MS2 were performed in a BSL-2 facility.

### 2.5. Host Cell Toxicity Test

Preliminary studies were performed to determine potential toxicity of the chemical disinfectants in the host cells. To evaluate the chemical-induced toxicity of the disinfectants, hard water was used to prepare the inoculum instead of the virus. Hard water was mixed with each test concentration of the disinfectants in the ratio 1:1 and incubated at 4 °C for 30 min, while being vortexed every 10 min. At the end of reaction time, the test samples were immediately mixed with an equal amount of each neutralizing medium. Subsequently, serial ten-fold dilution (up to 10^−2^) was performed in the 2% FBS-supplemented medium of each cell or culture broth of *E. coli*. The subsequent procedures were conducted in the same manner as described above. The toxicity controls were examined for any cell death or inactivation of *E. coli* due to the presence of residual toxicity from the chemical used. Results were considered valid if no toxicity was observed at a 1:10 dilution.

### 2.6. Data Analysis

Experiments were performed in triplicates for each concentration of the disinfectant-virus mixtures. Inactivation efficacy against the viruses was expressed as the titer reduction in log units, presented as the difference between the virus titer after reaction with disinfectant and the control virus titer. The final virus titer was determined as the median value of the triplicates, and the minimal virucidal concentrations (MVC) indicating the minimum concentration required for inactivation of initial viral titer of ≥4 log, was determined. The virus inactivation was considered to be effective when the titer reduction was ≥4 log [7]. The average of three reduction values was calculated, and the standard deviation and median values were determined.

## 3. Results

The concentrations of the tested disinfectants were chosen based on their formulations of disinfectants that have a proven efficacy and an official approval from APQA against FMDV, to observe the point at which each of the test treatments produced efficient virus inactivation [6,7,8,14]. Subsequently, based on the outcome of the effective testing of disinfectant concentrations against FMDV, a sequential change of concentrations was applied in treatments against MS2 and ECBO. A summary of the results for the active substances tested against FMDV, MS2, and ECBO is shown in Table 1, Table 2 and Table 3. For each active substance, the range of test concentrations includes the recommended concentration used in FMDV commercial disinfectants, and a minimum of one active and one inactive concentration.

Concentration-dependent virucidal activity is observed for all the active substances. Their MVCs against FMDV, which indicate the lowest concentration required for inactivation of initial viral titer of ≥4 log, are as follows: 0.3 g/L PPMS; 1 g/L NaDCC; 0.5 g/L GLT; 0.25 g/L CA; and 0.2 g/L MA (Table 1). None of the concentrations of the active substance exhibits cytotoxicity at 1/10 and 1/100 dilutions. However, BZK at 2 g/L or lower concentrations does not exhibit virucidal efficacy against FMDV, and its higher concentration leads to cytotoxic reactions.

Similarly, in the case of MS2, the MVCs of the active substances are as follows: 8 g/L PPMS; 2 g/L NaDCC; and 8 g/L GLT (Table 2). However, CA and MA do not exhibit virucidal activity against MS2 at much higher concentrations than those against FMDV. Incidentally, BZK at 2 g/L does not exhibit virucidal efficacy against MS2, and 1/10 dilution of the 4 g/L treatments are toxic to the host bacteria. This property of BZK prevented proper detection of infectivity since the decrease in the infectivity titer of 4 log could not be detected.

The MVCs of PPMS and NaDCC against ECBO are 8 g/L and 2 g/L, respectively (Table 3). For GLT and BZK, dilutions ≥1/10 of ≥1 g/L treatments are cytotoxic for the Vero cells, and lower concentrations do not exhibit virucidal efficacy against ECBO. None of the concentrations of the CA and MA, which are much higher concentrations than those against FMDV, exhibit virucidal efficacy.

Excluding the active substances that failed to determine the MVCs against each virus, these substances exhibit virucidal activity against MS2 and ECBO at much higher concentrations than those against FMDV. The MVC of PPMS and NaDCC against FMDV is 0.3 g/L and 1 g/L, respectively; however, these concentrations do not exhibit sufficient virucidal efficacy against MS2 and ECBO, and a ≥4 log reduction is obtained with 8 g/L in both (double concentration of NaDCC). The MVC of GLT against FMDV is 0.5 g/L; however, this concentration does not exhibit sufficient virucidal efficacy against MS2, and a ≥4 log reduction is obtained with 8 g/L. The overall results from the comparison of the median reduction titer at different concentrations of the active substances are displayed in Figure 1.

All disinfectant products were found to be effective against FMDV at the authorized concentrations, thereby confirming that the active substances are functional against FMDV at these particular concentrations (Table 4). Thereafter, we test the efficacy of the disinfectants at these concentrations against MS2 and ECBO (Table 4). When PPMS-based products, disinfectants 1, 2, and 3, are used against MS2 and ECBO at the authorized concentrations, their virucidal activities are not enough to meet the 4-log reduction; hence, the dilution of the three disinfectants to 0.96% causes a ≥4.00 log reduction against MS2. However, the concentration is not enough to meet the 4-log reduction against ECBO, and a 1.60% adjustment of dilution to higher concentration that reach the effective concentrations of active substances against ECBO (8.00 g/L PPMS, 1.60 g/L MA, and 2.40 g/L SDBS) leads to limited determination of virus titer reduction due to cytotoxic reactions. Disinfectant 4, which contains the effective concentration of NaDCC against both MS2 and ECBO, appears to have a considerable virucidal efficacy against MS2 and ECBO at the authorized concentration; however, upon decreasing the concentration of the active substance to the lower concentration, such as 1.6 g/L NaDCC for disinfectant 4, there is a ≤4.00 log reduction against MS2 and ECBO. Furthermore, disinfectant 5 is effective against ECBO at the authorized concentration containing 1.67 g/L NaDCC, which is almost effective concentration. Although the log reduction in response to disinfectant 5 is limited to 2.80 at the authorized concentration of 0.33% (1.67 g/L NaDCC, 0.80 g/L AA, 0.74 g/L SBC, and 0.13 g/L SCA) against MS2, a 6.80 log reduction is obtained with a 0.40% increase in the concentration of the disinfectant (2.00 g/L NaDCC, 0.96 g/L AA, 0.88 g/L SBC, and 0.16 g/L SCA). Incidentally, disinfectant 6, 7, and 8, do not exhibit the virucidal efficacy necessary to meet the 4-log reduction against MS2 and ECBO at the authorized concentrations; hence, an adjustment of dilution to higher concentrations cause ≥ 4.00 log reductions against MS2. For ECBO, the virucidal activities of the three QAC/GLT-based products are not enough to meet the 4-log reduction or lead to cytotoxic reactions at higher concentration than the authorized concentration.

## 4. Discussion

FMDV is classified as one of the most important infectious animal disease agents based on its pathogenicity, the difficulty in eliminating it from the environment, and the significant economic impact associated with reported outbreaks [12]. As a preventive measure against FMD outbreaks, regulated FMD vaccination must be implemented. However, in the farming-dominated areas of South Korea, there is a low positivity rate of vaccine antibodies because of the inappropriate method of vaccination [17]. Therefore, complementing vaccination with a strict biosecurity strategy, such as disinfection, which prevents the increased risk of virus exposure and introduction in the farm, can be an effective control measure [6].

This study focused on efficiently performing the disinfectant approval procedures for FMDV, along with selecting its surrogate. Candidates were selected based on safety, ease of culturing and handling, and lower BSL level requirement than FMDV. We selected MS2 and ECBO for this study. The bacteriophage MS2 (MS2) is a small, non-enveloped RNA coliphage that infects *Escherichia coli*, and belongs to the family *Leviviridae* and genus *Levivirus* [12,18]. It is commonly planned for use as a surrogate of human enteric viruses and FMDV in extensive research for efficacy testing of virucidal decontaminants [12,15,18,19,20,21,22,23]. MS2 has been reported to exhibit sensitivity [22,24] for selected disinfectants recommended for FMDV decontamination [25,26]. Within the virus family *Picornaviridae* members of the genus *Enterovirus* are small, non-enveloped single-stranded RNA viruses, which are common pathogens and display high resistance to harsh environments, therefore included in virus disinfection research [9]. Among them, bovine enterovirus type 1 (ECBO) is a typical representative of this group of viruses [9,27]. In the veterinary areas, this virus is used as a reference of enveloped and/or non-enveloped viruses, including FMDV, for assessing the virucidal potential of disinfectants in the German Veterinary Society (DVG) guidelines and European standards [28,29,30,31]. To determine whether MS2 or ECBO can be a potential surrogate candidate according to the APQA testing guidelines, we examined and compared their responses with that of FMDV for susceptibility to disinfectants.

The virucidal efficacy testing of six active substances revealed that their effective concentrations against the two surrogate candidate viruses are much higher than those against FMDV in two or three instances: MS2 and ECBO were both more resistant to the active substances when evaluating the efficacies at these levels without any toxicity. Moreover, the virucidal efficacy became better with higher concentration of available active substance.

Our study confirmed that all the eight disinfectants that had been approved for use against FMDV are effective: the concentrations of PPMS in disinfectants 1, 2, and 3, and that of NaDCC in disinfectants 4 and 5 are within the effective concentration ranges of the respective active substances. Incidentally, FMDV is considered a category B virus, in terms of resistance to chemical agents, and is best inactivated with hypochlorites, alkalis, Virkon^®^, glutaraldehyde, and acids [32]. It is reported to be relatively resistant to detergents and other common disinfectants such as quaternary ammonium compounds [12,32]. In disinfectant 6 and 7, GLT is present in a concentration of >0.5 g/L, which is sufficient to make it effective against FMDV. In disinfectant 8, although the concentration of GLT is lower than its effective concentration, it is effective due to its formulation with another active substance, namely formaldehyde (FAL). Previous studies reported that FAL is a powerful broad-spectrum high-level disinfectant with potent viral and bacterial inactivation capabilities [33,34,35]. Hence, it is possible that the synergistic effect with GLT increases the virucidal efficacy of the entire disinfectant.

The virucidal activities at the approved concentrations of FMDV disinfectants against the candidate viruses were measured to compare their efficacy against FMDV. Surrogate candidate viruses, MS2 and ECBO, were tested at FMDV-approved concentrations of multiple commercially available disinfectants, many of which failed the 4-log reduction. The two viruses display higher levels of resistance against disinfectants as compared to FMDV. PPMS/MA-based products, disinfectant 1, 2, and 3, are not effective against MS2 and ECBO at their respective authorized concentrations. However, upon increasing the specific concentration of active substances to their respective nearly effective concentration, such as PPMS > 4.00 g/L, there is a ≥4 log reduction against MS2. However, for ECBO, increasing the PPMS concentration to its respective effective concentration (8.00 g/L) induces a cytotoxic effect. Reportedly, a commercial disinfectant based on the PPMS/MA mixture is fast-acting and stable under the organic soiling conditions, but has higher cytotoxicity than other peroxyacid mixtures, thereby limiting its proper efficacy assessment [36]. While the dilution of the disinfectant was adjusted to achieve the effective concentration of the targeted active substances, the overall increase in the concentrations of PPMS/MA mixture seems to cause toxicity of the disinfectants. Nevertheless, ECBO titer reductions by ≥2.80 at 8.00 g/L PPMS are observed, suggesting its effectiveness. Additionally, the use of undiluted virus inoculum with high initial titer (modified method application) for an overall increase in viral titers confirmed the efficacy of these disinfectants against ECBO; for instance, 7.00 log reduction for disinfectant 1, 8.00 log for 2, and 7.00 log for 3. Therefore, a higher concentration of disinfectant used is required against MS2 and ECBO, as compared to that against FMDV, to provide the same or higher disinfectant efficacy. Disinfectants 4 and 5, which contain NaDCC as the predominant active substance, are effective against MS2 and ECBO at similar concentrations as the authorized levels. The slight difference in the results of the virus titer reduction of disinfectant 4 may be due to the different approaches that were used in determining the infectivity, i.e., CPE for FMDV and ECBO, and plaque assay for MS2 and/or the difference in initial viral titers [6,37]. The effective concentration of the disinfectant 5 against MS2 is slightly higher than that against FMDV, thereby indicating that MS2 is more resistant to disinfection than FMDV.

Quaternary ammonium compounds (QAC) are used widely as disinfectants but are reported to be less effective against non-enveloped viruses [5,38]. Similar to our results, previous studies had reported ineffectiveness of QAC against FMDV in a suspension test [5,39]; however, a combination of QAC and GLT was reported to exhibit effectiveness against FMDV [5]. QAC/GLT-based products, disinfectant 6, 7, and 8 are thoroughly ineffective against both MS2 and ECBO at the authorized concentrations. However, increasing the overall concentrations of disinfectant 2-fold, resulting ≥ 1.34 g/L GLT, is effective against MS2; however, for disinfectant 8 containing FAL, 1.5-fold increase in overall concentration is ineffective, but 3-fold increase has been confirmed to be effective against MS2 with high initial titer inoculum despite the cytotoxicity. Although the concentration of the active substance, GLT, is lower than its effective concentration, this disinfectant is effective due to its complex formulation nature. As mentioned above, FAL is a high-level disinfectant that effectively and rapidly inactivates many different types of viruses [33,35], and it is possible that the synergistic effect of the main substances increases the virucidal efficacy of the entire disinfectant. Gosling et al. (2017) reported that there is an additive effect upon combining with the GLT/QAC-based product and cited that FAL-based products are more effective as disinfectants than GLT/BZK-based products [34]. Moreover, previous studies had reported that the products containing a mixture of FAL and GLT/QAC performed significantly better in reducing *Salmonella* and avian influenza virus [33,34,40]. However, FAL and GLT/QAC formulations have high cytotoxicity [8,41], which does not permit proper assessment. For ECBO, using the QAC/GLT-based products used at more than twice the authorized concentrations (2.00%) induce cytotoxic effects. While the dilution of the disinfectant was adjusted to achieve the effective concentration, the increase in the concentrations of GLT (1.00 g/L), which had expressed cytotoxicity during individual active substance testing against ECBO, appears to have caused toxicity of the disinfectants.

The U.S. Environmental Protection Agency (EPA) allows surrogate viruses to be used for substantiating virucidal activity not only for difficult-to-culture pathogenic viruses but also for new and emerging viruses [4]. The approach is based on the scientific rationale that if disinfectants inactivate more challenging surrogates than specific target organisms, then they should be expected to inactivate the target organism [42]. With this approach, EPA ranks viral pathogens with respect to their tolerance or resistance to chemical disinfectants and divides the viruses into three subgroups based on size. Most susceptible to most resistant tiers of viruses are: enveloped, large (50–100 nm) non-enveloped, and small (<50 nm) enveloped viruses [4,43]. Incidentally, many studies identified that non-enveloped viruses are more resistant to inactivation than enveloped viruses, though they can vary depending on the virus type and disinfectant formulation [9,44]. Moreover, a recent study has reported that the surrogate virus is more resistant to the applied chemical disinfectants than the targeted outbreak organism [45]. Consistently, our observations show that MS2 and ECBO are more susceptible to higher concentrations of disinfectants, as compared to FMDV. Based on our results, MS2 and ECBO are both found to be more resistant to disinfection. This information can confirm the hypothesis that the two viruses are appropriate surrogates for FMDV for testing the efficacy of virucidal disinfectants, according to the method described by the APQA guidelines.

In conclusion, the use of MS2 and ECBO as surrogate viruses facilitate the experiments under BSL-2 facility conditions, thereby avoiding high-level containment areas, which are legally necessary for handling FMDV in South Korea. Based on our research, we anticipate professionals and advisory groups making the most appropriate decision for selecting an optimal surrogate virus against FMDV when updating the standard testing methods and time-effective system requirements for the approval of chemical disinfectants in South Korea.

## Figures and Tables

**Figure 1 viruses-14-02590-f001:**
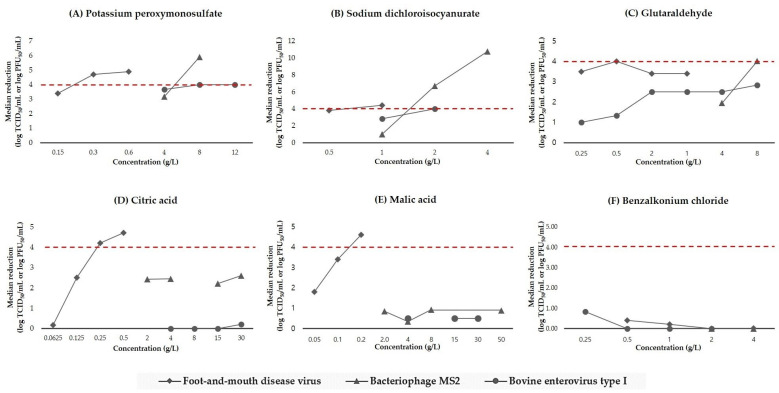
Comparison of the median reduction titer (in log TCID_50_/mL or PFU/mL) at concentrations of the active substances against foot-and-mouth disease virus (square solid line), bacteriophage MS2 (triangle solid line), and bovine enterovirus type 1 (circle solid line). (**A**) Potassium peroxymonosulfate. (**B**) Sodium dichloroisocyanurate. (**C**) Glutaraldehyde. (**D**) Citric acid. (**E**) Malic acid. (**F**) Benzalkonium chloride. The red dashed line indicates virucidal effect threshold.

**Table 1 viruses-14-02590-t001:** Virucidal activity of active substances against foot-and-mouth disease virus.

Disinfectant (Active Substance)	Concentration (g/L)	Log_10_ Reduction							Dilution with Toxicity
		E1	E2	E3	Mean (±SD)			Median	
Potassium peroxymonosulfate (PPMS)	0.15	3.80	3.40	0.20	2.47	±	1.97	3.40	
	0.3	**4.70**	**4.90**	**4.40**	**4.67**	**±**	**0.25**	**4.70**	
	0.6	**4.80**	**4.90**	**4.90**	**4.87**	**±**	**0.06**	**4.90**	
Sodium dichloroisocyanurate (NaDCC)	0.5	3.80	3.80	3.60	3.73	±	0.12	3.80	
	1	**4.40**	**4.40**	**4.60**	**4.47**	**±**	**0.12**	**4.40**	
Glutaraldehyde (GLT)	0.25	**4.00**	3.17	3.50	3.56	±	0.42	3.50	
	0.5	3.83	**4.17**	**4.00**	**4.00**	**±**	**0.17**	**4.00**	
	1 *	3.40	3.40	3.60	3.47	±	0.12	3.40	10^−1^
	2 *	3.40	3.40	3.40	3.40	±	0.00	3.40	10^−1^
Citric acid (CA)	0.0625	0.17	0.17	0.17	0.17	±	0.00	0.17	
	0.125	2.00	2.50	2.50	2.33	±	0.29	2.50	
	0.25	**4.20**	**4.20**	**4.90**	**4.43**	**±**	**0.40**	**4.20**	
	0.5	**4.10**	**4.70**	**4.80**	**4.53**	**±**	**0.38**	**4.70**	
Malic acid (MA)	0.05	1.80	1.80	1.60	1.73	±	0.12	1.80	
	0.1	3.00	3.40	3.40	3.27	±	0.23	3.40	
	0.2	**4.60**	**4.30**	**4.80**	**4.57**	**±**	**0.25**	**4.60**	
Benzalkonium chloride (BZK)	0.5	1.00	0.40	0.00	0.47	±	0.50	0.40	
	1	0.00	0.20	1.20	0.47	±	0.64	0.20	
	2	0.20	0.00	0.00	0.07	±	0.12	0.00	
	4 *	0.00	0.00	0.00	0.00	±	0.00	0.00	10^−1^, 10^−2^

Values in bold indicate effective virucidal activity (viral titer ≥ 4 log_10_). E, experiment; SD, standard deviation. * Limitation encountered while calculating owing to cytotoxicity.

**Table 2 viruses-14-02590-t002:** Virucidal activity of active substances against bacteriophage MS2.

Disinfectant (Active Substance)	Concentration (g/L)	Log_10_ Reduction							Dilution with Toxicity
		E1	E2	E3	Mean (±SD)			Median	
Potassium peroxymonosulfate (PPMS)	4	3.11	3.16	4.02	3.43	±	0.51	3.16	
	8	**4.26**	**5.89**	**5.91**	**5.35**	**±**	**0.95**	5.89	
Sodium dichloroisocyanurate (NaDCC)	1	1.27	0.99	1.00	1.09	±	0.16	1.00	
	2	**7.67**	**6.57**	**6.70**	**6.98**	**±**	**0.60**	6.70	
	4	**10.87**	**10.78**	**10.58**	**10.74**	**±**	**0.15**	10.78	
Glutaraldehyde (GLT)	4	1.16	1.94	2.44	1.85	±	0.65	1.94	
	8	**4.20**	**4.00**	**3.84**	**4.01**	**±**	**0.18**	4.00	
Citric acid (CA)	2	2.42	2.29	2.94	2.55	±	0.34	2.42	
	4	2.85	2.36	2.45	2.55	±	0.26	2.45	
	15	2.22	2.10	2.25	2.19	±	0.08	2.22	
	30	2.72	2.60	2.47	2.60	±	0.13	2.60	
Malic acid (MA)	2	0.21	0.85	0.92	0.66	±	0.39	0.85	
	4	0.79	0.18	0.34	0.44	±	0.32	0.34	
	8	1.29	0.92	0.85	1.02	±	0.24	0.92	
	50	0.85	0.92	0.89	0.89	±	0.04	0.89	
Benzalkonium chloride (BZK)	2	0.00	0.52	0.00	0.17	±	0.30	0.00	
	4 *	0.00	0.00	0.00	0.00	±	0.00	0.00	10^−1^

Values in bold indicate effective virucidal activity (viral titer ≥ 4 log_10_). E, experiment; SD, standard deviation. * Limitation encountered while calculating owing to cytotoxicity.

**Table 3 viruses-14-02590-t003:** Virucidal activity of active substances against bovine enterovirus type 1.

Disinfectant (Active Substance)	Concentration (g/L)	Log_10_ Reduction							Dilution with Toxicity
		E1	E2	E3	Mean (±SD)			Median	
Potassium peroxymonosulfate (PPMS)	4	3.67	3.17	3.67	3.50	±	0.29	3.67	
	8	**4.67**	**4.17**	**4.17**	**4.34**	**±**	**0.29**	4.17	
	12	**4.83**	**4.67**	**4.17**	**4.56**	**±**	**0.34**	4.67	
Sodium dichloroisocyanurate (NaDCC)	1	3.83	3.00	2.83	3.22	±	0.54	3.00	
	2	**4.00**	**4.00**	**4.17**	**4.06**	**±**	**0.10**	4.00	
Glutaraldehyde (GLT)	0.25	1.17	0.67	1.00	0.95	±	0.25	1.00	
	0.5	1.33	1.00	1.33	1.22	±	0.19	1.33	
	1 *	2.50	2.50	2.50	2.50	±	0.00	2.50	10^−1^, 10^−2^
	2 *	2.50	2.50	2.50	2.50	±	0.00	2.50	10^−1^, 10^−2^
	4 *	2.50	2.33	3.70	2.84	±	0.75	2.50	10^−1^, 10^−2^
	8 *	2.50	2.83	2.83	2.72	±	0.19	2.83	10^−1^, 10^−2^
Citric acid (CA)	4	0.00	0.00	0.00	0.00	±	0.00	0.00	
	8	0.00	0.50	0.00	0.17	±	0.29	0.00	
	15	0.50	0.00	0.00	0.17	±	0.29	0.00	
	30	0.80	0.20	0.00	0.33	±	0.42	0.20	
Malic acid (MA)	4	0.50	0.00	0.50	0.33	±	0.29	0.50	
	15	1.00	0.50	0.50	0.67	±	0.29	0.50	
	30	1.20	0.20	0.50	0.63	±	0.51	0.50	
Benzalkonium chloride (BZK)	0.25	0.83	0.33	0.83	0.66	±	0.29	0.83	
	0.5	0.00	0.00	0.33	0.11	±	0.19	0.00	
	1 *	0.00	0.00	0.00	0.00		0.00	0.00	10^−1^
	2 *	0.00	0.00	0.00	0.00	±	0.00	0.00	10^−1^

Values in bold indicate effective virucidal activity (viral titer ≥ 4 log_10_). E, experiment; SD, standard deviation. * Limitation encountered during determination owing to cytotoxicity.

**Table 4 viruses-14-02590-t004:** Description and virucidal activity of commercial disinfectants tested against foot-and-mouth disease virus, bacteriophage MS2, and bovine enterovirus type 1.

Disinfectant	Active Substance	Concentration of Disinfectant (%)	Active Substance Concentration at Use Concentration (g/L)	Log_10_ Reduction		
				FMDV	MS2	ECBO
1	PPMS + MA	**0.04**	0.20 + 0.04	5.30	1.50	0.50
		0.48	2.40 + 0.48	-	2.10	2.00
		0.96	4.81 + 0.96	-	4.00	2.00
		1.60	8.00 + 1.60	-	-	≥3.00 *
2	PPMS + MA + SDBS	**0.07**	0.33 + 0.07 + 0.10	4.70	0.20	1.30
		0.12	0.61 + 0.12 + 0.18	-	0.30	1.33
		0.24	1.22 + 0.24 + 0.37	-	1.30	1.67
		0.48	2.40 + 0.48 + 0.72	-	2.80	1.67
		0.96	4.81 + 0.96 + 1.44	-	4.80	≥3.33 *
		1.60	8.00 + 1.60 + 2.40	-	-	≥3.00 *
3	PPMS + MA + SDBS	**0.08**	0.38 + 0.08 + 0.12	4.60	2.10	1.20
		0.48	2.40 + 0.48 + 0.72	-	1.90	2.17
		0.96	4.81 + 0.96 + 1.44	-	4.00	3.67
		1.60	8.00 + 1.60 + 2.40	-	-	≥2.80 *
4	NaDCC	0.42	1.60	4.33	3.80	3.29
		**0.53**	2.02	4.20	5.20	4.80
5	NaDCC + AA + SBC + SCA	0.27	1.33 + 0.64 + 0.59 + 0.11	4.33	1.26	3.20
		**0.33**	1.67 + 0.80 + 0.74 + 0.13	4.40	2.80	4.80
		0.40	2.00 + 0.96 + 0.88 + 0.16	-	6.80	4.30
6	QAC (BZK + DDAC) + GLT	**0.63**	1.55 (1.07 + 0.49) + 0.67	4.00	0.00	2.80
		1.25	3.11 (2.13 + 0.98) + 1.34	-	7.80	≥0.00 *
		2.50	6.22 (4.27 + 1.95) + 2.68	-	12.20^*^	≥1.17^*^
		7.46	18.55 (12.73 + 5.82) + 8.00	-	-	≥0.17 *
7	QAC (BZK) + GLT	**0.50**	0.50 + 0.75	4.00	0.00	1.70
		1.00	1.00 + 1.50	-	6.39	2.00
		2.00	2.00 + 3.00	-	8.10	2.17
		5.56	5.56 + 8.33	-	-	≥2.50
8	QAC (BZK) + GLT + FAL	**0.67**	0.40 + 0.33 + 0.53	4.00	0.00	0.50
		1.00	0.60 + 0.50 + 0.80	-	0.00	0.67
		2.00	1.20 + 1.00 + 1.60	-	4.20 *	≥1.17 *
		16.03	9.62 + 8.01 + 12.79	-	-	≥0.00 *

PPMS, potassium peroxymonosulfate; NaDCC, sodium dichloroisocyanurate; GLT, glutaraldehyde; MA, malic acid; BZK, benzalkonium chloride; QAC, quaternary ammonium chloride; DDAC, didecyldimethylammonium chloride; SDBS, sodium dodecylbenzene sulfonate; AA, adipic acid; SBC, sodium bicarbonate; SCA, sodium carbonate; FAL, formaldehyde. FMD, foot-and-mouth disease virus; MS2, bacteriophage MS2; ECBO, bovine enterovirus type 1; -, not done. Bold type, authorized concentrations used against foot-and-mouth disease virus. * Limitation encountered while calculating owing to cytotoxicity.

## Data Availability

Not applicable.

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
