# Peer review of "Surrogate Selection for Foot-and-Mouth Disease Virus in Disinfectant Efficacy Tests by Simultaneous Comparison of Bacteriophage MS2 and Bovine Enterovirus Type 1"

_viruses, 2022, doi:10.3390/v14122590_

Round 1

Reviewer 1 Report

In this study, Rhee et al. presented the possible candidate surrogates, MS2 and ECBO for FMDV in future use in testing various disinfectants against FMDV. The study looks incomplete. The authors just used the above-mentioned two surrogate candidate viruses but did not use MS2 or ECBO carrying FMDV, which is highly recommended to come across the study's conclusion. Once they construct the MS2 or ECBO surrogate carrying FMDV, they should test the various disinfectants and conclude the study. I recommend that the authors complete their research design to translate their idea to reality to use FMDV in candidate MS2 or ECBO surrogate viruses and be able to perform FMDV-related experiments in a BSL2 laboratory.  

L42-43 modify the sentence.

L49 Conducted instead of condeucted

Please add a sentence to the introduction that FMDV-related experiments should be conducted in a BSL-3 facility.

L283 Selected not select

As shown in the results, the viricidal activity of some of the disinfectants against MS2 or ECBO and FMDV is different. Therefore, how can the authors use these surrogates for FMDV to test different disinfectants to restrict FMDV through biosecurity measures?

The authors just used MS2 and ECBO to see the viricidal activity of different disinfectants. To prove that MS2 or ECBO could be used as an FMDV surrogate, it is highly recommended to introduce FMDV in these surrogates and test the various disinfectant viricidal efficiency.

Reviewer 2 Report

This paper describes a laboratory comparison of microorganism resistance to numerous disinfectants, for the purposes of establishing surrogacy of non-pathogenic microorganisms for pathogenic FMDV.

Comments;

Lines 82 – 85 – First mention of these chemical groups should be spelled out. I.e., PPMS  should be named as potassium peroxymonosulfate (PPMS).

Were assays performed to demonstrate neutralizer efficacy? i.e., Virus spiked into neutralized disinfectant to demonstrate complete neutralization. This is required for ASTM E2197. Demonstration of effective and instant neutralization of test chemistries, for virucidal activity, is important, in addition to the demonstration of no cytotoxic effects on cell lines.

Line 151 – since coliphage particle to infectious unit is 1 (rather than ~0.69 for mammalian virus), why express recovery and reductions in PFU50? Why not PFU, using more simple and direct calculations? Was this to have better comparison to the mammalian virus units? Perhaps state this reasoning if so.

Line 178 – is there a citation for the ‘formulations with proven efficacy against FMDV”.?  

Tables 1,2,3 – it would be nice to have the active substance abbreviation alongside the full name, to cross reference with the text.

Figure 1 – why are the x-axis values in descending order, from left to right, with regards to concentration of active? Seems more intuitive to have the x-axis values increasing from left to right, but perhaps there is a reason for this the authors can explain?

Line 234 – Sentence starting “Thereafter,” is awkwardly worded, consider revision.

Line 243-244 – Disinfectant 4 seems more effective against the surrogates than FMDV, which draws into question their validity as surrogates for that active chemistry. Or were these results due to differences in titer, may need to explain this result as it differs from the other disinfectants and has implications for the suitability of the surrogates.

Line 319 – need to caveat that Disinfectant 4 is the exception to the notion that candidate surrogate resistance was higher than FMDV

Line 339 – again here is an opportunity to mention the Disinfectant 4 was more effective against the surrogates, but the authors do not disclose this. Please add at some point in this paper.

Round 2

Reviewer 1 Report

The authors response was satisfactory.